# Carbohydrate-Responsive Element-Binding Protein-Associated Metabolic Changes in Chemically Induced Hepatocarcinogenesis Mouse Model

**DOI:** 10.3390/ijms26146932

**Published:** 2025-07-18

**Authors:** Maren Engeler, Majedul Karim, Marcel Gischke, Franziska Willer, Helen Leiner, Jessica Prey, Paul Friedrich Ziegler, Frank Dombrowski, Silvia Ribback

**Affiliations:** Institute for Pathology, University Medicine Greifswald, 17475 Greifswald, Germany; maren.engeler@med.uni-greifswald.de (M.E.); smmajedul.karim@med.uni-greifswald.de (M.K.); marcel.gischke@med.uni-greifswald.de (M.G.); franzi.willer@web.de (F.W.); f.leiner@t-online.de (H.L.); jessica.prey@med.uni-greifswald.de (J.P.); paulfriedrich.ziegler@med.uni-greifswald.de (P.F.Z.); frank.dombrowski@med.uni-greifswald.de (F.D.)

**Keywords:** ChREBP, PI3K/AKT/mTOR, diethylnitrosamine, chemically induced hepatocarcinogenesis, cancer metabolism, hepatocellular carcinoma

## Abstract

The Carbohydrate-Responsive Element-Binding Protein (ChREBP) is a glucose-sensitive transcription factor that regulates the carbohydrate and lipid metabolism. We investigated its cell-type-specific role in hepatocarcinogenesis using a chemically induced mouse model. Additionally, we examined the functions of its isoforms, ChREBPα and ChREBPβ. After the diethylnitrosamine (DEN) administration, we analyzed hepatocellular adenomas and carcinomas in systemic ChREBP-knockout (KO), hepatocyte-specific ChREBP-KO (L-KO), and wildtype (WT) mice at 4, 12, and 36 weeks using histology, morphometry, proliferation measurements, immunohistochemistry, a Western blot, and a quantitative PCR. Tumors developed 36 weeks after the DEN administration in 27% of WT mice but less frequently in KO (18%) and L-KO (9%) mice. However, preneoplastic foci were less common in KO mice but not in L-KO mice (39% vs. 9%; *p* < 0.05). L-KO hepatocytes exhibited lower proliferation, while KO tumors showed the downregulation of AKT/mTOR signaling, glycolysis, and lipogenesis compared to WT tumors. Our results showed that the liver-specific loss of ChREBPα, while ChREBPβ remained active, significantly reduced the tumor progression, suggesting an oncogenic role for ChREBPα. In contrast, the systemic knockout of both ChREBPα and ChREBPβ reduced the tumor initiation but did slightly prevent tumor progression, indicating that ChREBPβ may exert tumor-suppressive functions.

## 1. Introduction

Hepatocellular carcinoma (HCC) is a highly prevalent primary liver cancer globally, with more than 850,000 new cases diagnosed every year. It accounts for 75–85% of primary liver cancers and is responsible for around 750,000 deaths annually, highlighting its aggressive nature [1]. As a result, HCC remains the third leading cause of the cancer-related death worldwide [2]. The pathogenesis of HCC is a complex and multifactorial process, typically triggered by chronic liver diseases and advancing from DNA damage to dysplasia, adenoma formation, and the malignant transformation of hepatocytes [3]. The sexual dimorphism in HCC development is evident, whereas a male predisposition—regardless of the underlying etiology, such as viral hepatitis, alcoholic liver disease, obesity, diabetes, and non-alcoholic fatty liver disease—is known [4,5]. The increasing prevalence of obesity worldwide and the inevitable exposure to toxic compounds may point to a bleak future for the incidence of HCC. However, the understanding of underlying molecular mechanisms involved in the HCC development and progression is still limited.

Diethylnitrosamine (DEN) is a DNA alkylating agent widely used to study liver cancer and particularly HCC in rodent models [6]. N-nitroso compounds, particularly DEN, are known potent hepatic carcinogens that can cause DNA damage and mutations through metabolic alterations in the liver following a single dose [7]. It is present in tobacco smoke as well as food products, such as processed meat and alcoholic beverages [8]. Therefore, DEN is able to mimic human hepatocarcinogenesis, which ultimately results in the development of preneoplastic lesions, hepatocellular adenomas (HCAs), and HCC [9]. Upon administration, DEN is metabolized by cytochrome P450 enzymes in the liver to form alkylating agents, which can cause DNA strand breakage, mutagenic DNA adduct formation, and the activation of reactive oxygen species (ROS) [10,11]. The subsequent necrosis and regeneration of hepatocytes promotes mutation and neoplastic transformation, ultimately resulting in hepatocarcinogenesis [12]. Moreover, due to the high heterogeneity in the cellular morphology and genetic landscape of HCC, DEN-induced carcinogenesis can play an essential role in understanding the cellular and molecular processes involved in the carcinogenic transformation of hepatocytes [13].

Cancer cells undergo metabolic reprogramming to ensure adequate nutrition and support their continued growth and division [14]. In particular, cancer cells exhibit a high rate of glycolysis and nucleotide biosynthesis and an enhanced lipid and amino acid metabolism [14,15]. The transcription factor carbohydrate-responsive element-binding protein (ChREBP) plays a crucial role in the coordinated regulation of the carbohydrate and lipid metabolism in the liver by modulating the gene expression involved in glycolysis and de novo lipogenesis [16,17,18]. Recently, it has been reported that the upregulated hepatic ChREBP expression mediates the metabolic alteration during HCC progression [19,20]. Emerging evidence reveals that ChREBP enhances the activity of AKT/mTOR and Ras/MAPK signaling cascades to promote the growth, survival, and metabolism of altered hepatocytes [19]. Previous studies in mice from our lab reported that a ChREBP deficiency may play a protective role against HCC development [19]. We also provided evidence that a systemic ChREBP loss strongly delays or impairs hepatocarcinogenesis driven by AKT/c-Met overexpression in mice [20,21].

Two isoforms are described for ChREBP. ChREBPα (~95 kDa, 864 base pairs (bp)) is the commonly described glucose-dependent form, which is present in the cytosol and has its translational start site in exon 1a. ChREBPβ is translated from exon 4 and is therefore a truncated isoform (~72 kDa, 687 bp) which is consecutively active and localized in the nucleus due to the loss of the low glucose inhibitory domain and nuclear export signals. Both isoforms are expressed in the liver [22,23]. ChREBPβ regulates de novo lipogenesis more strongly than ChREBPα, as the expression of ChREBPβ in the human liver correlates positively with the expression of lipogenic enzymes (such as fatty acid synthase (FASN)), especially under diabetic or obese metabolic conditions [24].

Therefore, given its importance in hepatic metabolism, we characterized and investigated carcinogenesis in both hepatic and systemic ChREBP-knockout mice. In this context, the present study aimed to investigate the organ-specific pathogenic role of ChREBP isoforms in the DEN-induced hepatocarcinogenesis.

## 2. Results

The liver tissue from systemic ChREBP-knockout mice (KO), liver-ChREBP-knockout mice (L-KO), and C57Bl/6J wildtype mice (WT), each with and without a single administration of DEN, was used for the analyses. Histomorphologically, glycogen storage or a slight fatty degeneration of the hepatocytes was observed in certain animals.

### 2.1. Proliferative Activity of Unaltered Liver Tissue

The Ki-67 labeling index (Ki-67-LI) was used to determine the proliferation activity and corresponds to the percentage of Ki-67-positive cell nuclei of 2000 hepatocytes. Three representative fields of view were analyzed at 100× magnification (Figure 1A). Brown cell nuclei are in mitotic cell division and are counted to determine the proliferation activity (Figure 1B).

The DEN application led to a significantly higher proliferation in ChREBP-KO and liver-ChREBP-KO after 4 weeks than in the respective controls (KO DEN 4W vs. KO control 4W: Ki-67 LI 6.17 ± 1.31% (mean ± SEM) (*n* = 15) vs. 1.09 ± 0.35% (*n* = 17); *p* = 0.002; L-KO DEN 4W vs. L-KO control 4W: 2.02 ± 0.31% (*n* = 15) vs. 0.64 ± 0.13% (*n* = 22); *p* < 0.001). Interestingly, the liver-ChREBP-KO showed a markedly decreased proliferation compared to ChREBP-KO after 4 weeks with the DEN application, L-KO vs. KO 2.02 ± 0.31% (*n* = 15) vs. 6.17 ± 1.31% (*n* = 15); *p* = 0.009, and after 12 weeks without DEN: L-KO control vs. KO control 0.64 ± 0.13% (*n* = 22) vs. 1.09 ± 0.35% (*n* = 17); *p* = 0.009. Compared to the WT, the proliferation activity of liver-ChREBP-KO tended to be reduced (L-KO DEN 4W vs. WT DEN 4W: 2.02 ± 0.31% (*n* = 15) vs. 4.66 ± 2.19% (*n* = 14); n.s.) (Figure 1C).

### 2.2. Glycogen Storage in Hepatocytes

Cytoplasmic glycogen storage in liver tissue only occurred in ChREBP-KO mice (Appendix A). In all three observation periods, the glycogen storage in hepatocytes was significantly increased in the ChREBP-KO control (4 weeks: 8.6 ± 2.5% (*n* = 20)) compared to the liver-ChREBP-KO control (4 weeks: 0% (*n* = 25); *p* = 0.003) and the WT control (4 weeks: 0% (*n* = 24); *p* = 0.003). (Appendix A).

In addition, glycogen stores were also found in the cell nuclei of ChREBP-KO mice (Appendix A).

### 2.3. Progression of Preneoplastic Foci to Liver Adenomas and Carcinomas

Various lesions were detected in the liver tissue over a test period of 36 weeks (Table 1). Preneoplastic foci (Figure 2A,B) are characterized by small islands of morphologically altered hepatocytes with a basophilic or clear cytoplasm and without nuclear atypia and cell dysplasia. The hepatocyte cell beads appear irregularly shaped. These foci occurred significantly more frequently in the liver-ChREBP-KO DEN than in the ChREBP-KO DEN (39.1% (*n* = 23) vs. 9.09% (*n* = 22); *p* = 0.021). Furthermore, compared to WT DEN mice, significantly fewer preneoplastic lesions developed in ChREBP-KO DEN (WT DEN vs. KO DEN: 40.9% (*n* = 22) vs. 9.09% (*n* = 22); *p* = 0.021) (Table 1). HCAs (Figure 2C,D) measured 1–5 mm and showed a disturbed parenchymal architecture, mild nuclear atypia, and cell dysmorphism, such as increased fat or glycogen storage, but only a few mitoses and slightly increased proliferation. Larger (diameter 5–15 mm) and invasively growing neoplasms with incipient dedifferentiation, enlarged cell nuclei, and polyploidy were defined as HCC (Figure 2E,F and Figure 3). For statistical analysis, the HCA and HCC were combined under the term tumors.

Liver-ChREBP-KO (DEN: 8.7% of animals (*n* = 23)) and ChREBP-KO mice (DEN: 18.2% of animals (*n* = 22)) tended to develop fewer tumors than WT mice (DEN: 27.3% of animals (*n* = 22); n.s.). Liver-ChREBP-KO mice tended to show the fewest tumors and, strikingly, developed no HCC (Table 1).

### 2.4. Altered Activation of Glycolysis, Lipogenesis, PI3K/AKT/mTOR, and Ras/MAPK Pathways in Liver Tumors

To explore the effects of ChREBP in the development of DEN-induced liver tumors, we examined the protein expression of pyruvate kinase M2 (PKM2), hexokinase II (HK-2), acetyl-CoA carboxylase (ACAC), fatty acid synthase (FASN), phosphorylated/activated AKT (p-AKT), the phosphorylated mechanistic target of rapamycin (p-mTOR), phosphorylated eukaryotic translation initiation factor 4E binding protein 1 (p-4E-BP1), and phosphor-extracellular signal regulated kinase 1/2 (p-ERK1/2) proteins in tumor and non-tumor tissues from WT and ChREBP-KO mice by immunohistochemistry and the Western blot analysis. In liver tumors, the decreased expression of glycolytic enzymes PKM2 and HK-2 contributed to the enhanced proliferation of hepatocytes in liver tumors. The energy demands of the altered hepatocytes were met by the increased expression of the de novo lipogenesis enzymes ACAC and FASN. The expression of p-AKT, p-mTOR, and p-4E-BP1 in tumorous liver tissues was notably lower in ChREBP-KO compared to WT mice. Furthermore, we evaluated the expression of the Ras/MAPK candidate p-ERK1/2, which also showed a decreased expression in liver tumors of ChREBP-KO mice compared to the WT (Figure 4). The densitometric quantification of these results supports the described observations (Appendix A).

In the comparison of the WT tumor tissue (indicated by * in Figure 5A) with the unmodified liver tissue, the tumor showed a higher expression of HK-2, PKM2, ACAC, and FASN and a subsequent upregulation of glycolysis and de novo lipogenesis. In addition, p-AKT/p-mTOR, including the downstream effector p-4E-BP1 and the Ras/MAPK candidate p-ERK1/2, was increasingly expressed in the WT tumor.

In the ChREBP-KO tumor tissue (indicated by * in Figure 5B), glycolysis and especially de novo lipogenesis were significantly upregulated compared to the unaltered liver tissue, indicated by the higher expression of HK-2, PKM2, ACAC, and FASN. Furthermore, there was a slight downregulation of p-AKT and p-mTOR and a clearly reduced expression of p-4E-BP1 and p-ERK1/2 in ChREBP-KO tumors compared to the adjacent liver tissue.

These results provide evidence that the activation of ChREBP-associated metabolic pathways is essential for regulating the altered hepatocyte proliferation and thus the progression of hepatocarcinogenesis.

### 2.5. Protein Expression of Unaltered Liver Tissue

The protein expression in the non-altered liver tissue was analyzed across all experimental groups using Western blotting and immunohistochemical staining. Hepatocytes from ChREBP-KO mice treated with DEN showed, in several cases, a significant upregulation of the AKT/mTOR pathway, including the downstream effector p-4E-BP1, compared to WT mice. Similarly, glycolysis—indicated by the significantly increased expression of HK-2—was upregulated in ChREBP-KO mice compared to WT mice (Appendix A).

In liver-ChREBP-KO mice, the expression of the glycolytic enzyme PKM2 was slightly elevated, while the lipogenic enzymes ACAC and FASN were partly significantly increased, particularly following the DEN treatment. The AKT/mTOR pathway was significantly upregulated in these mice compared to the WT liver tissue, whereas no differences were observed for the p-ERK1/2 expression (Appendix A).

Western blot results were further quantified and validated via a densitometric analysis (Appendix A). A graphical representation of the immunohistochemically stained area is shown in Appendix A.

These findings were corroborated by the gene expression analysis in the unaltered liver tissue using qPCRs. Upregulated de novo lipogenesis resulted in significantly more FASN in liver-ChREBP-KO control mice than in ChREBP-KO control mice. Glycolysis, indicated by PKM2, tends to be upregulated in ChREBP-KO mice, especially after the DEN application, compared to WT mice (Appendix A).

### 2.6. Body Weight Progression and Liver-to-Body-Weight Ratio

It is generally noticeable that the control mice of all genotypes showed a slight weight loss around the 28th week, which normalized after 4 weeks. The animals with the DEN application showed a constant weight progression (Appendix A).

Over the entire observation period, the ChREBP-KO mice gained the least weight, regardless of DEN applications. At all time points, the body weight of the KO DEN mice (36 weeks: 30.7 ± 0.4 g (*n* = 22)) was significantly lower than that of the L-KO DEN (36 weeks: 32.7 ± 0.5 g (*n* = 23); *p* = 0.005) or WT DEN mice (36 weeks: 32.7 ± 0.6 g (*n* = 22); *p* = 0.006). Similarly, KO control mice (36 weeks: 30.9 ± 0.6 g (*n* = 24)) consistently had a significantly lower body weight compared to the WT control (36 weeks: 32.7 ± 0.7 g (*n* = 25); *p* = 0.045). Compared to the L-KO control, the KO control mice had a lower body weight only during the first 20 weeks (20 weeks: 29.3 ± 0.5 g (*n* = 24) vs. 30.8 ± 0.3 g (*n* = 25); *p* = 0.007). From the 20th week onwards, the weight of the L-KO control (36 weeks: 31.0 ± 0.3 g (*n* = 25)) drops to the level of the KO mice and thus becomes significantly lower than that of the WT control (36 weeks: 32.7 ± 0.7 g (*n* = 25); *p* = 0.029) (Appendix A).

At all timepoints, the liver-to-body-weight ratio (in percentage) of the ChREBP-KO (DEN 36 weeks: 6.25 ± 0.23% (*n* = 22)) was significantly higher than that of the liver-ChREBP-KO (DEN 36 weeks: 4.89 ± 0.16% (*n* = 23); *p* < 0.001) or the WT mice (DEN 36 weeks: 4.54 ± 0.43% (*n* = 22); *p* = 0.002). The ratio of liver-ChREBP-KO mice was tendentially higher than the WT but only differed significantly after 4 weeks (DEN: 5.13 ± 0.10% (*n* = 17) vs. 4.59 ± 0.13% (*n* = 16); *p* = 0.004). There was no significant variation between the DEN administration and control groups, except at the beginning of the experiment after 4 weeks in the liver-ChREBP-KO (DEN vs. control; 5.13 ± 0.10% (*n* = 17) vs. 6.01 ± 0.19% (*n* = 25); *p* < 0.001) and WT mice (DEN vs. control; 4.59 ± 0.13% (*n* = 16) vs. 5.21 ± 0.20% (*n* = 24); *p* = 0.016) (Appendix A).

## 3. Discussion

In the present study, hepatocellular proliferation, metabolic and protooncogenic signaling cascades, and liver tumors were investigated to understand the effects of the transcription factor ChREBP on DEN-induced hepatocarcinogenesis. The DEN-induced mouse HCC model has shown the capability to mimic the HCC development observed in humans [9], which is crucial for understanding the diverse molecular pathways involved in tumor initiation and progression. Our results demonstrate that DEN-induced alterations in liver metabolism significantly contributed to the generation of preneoplastic foci and tumors. In this study, systemic ChREBP-KO, liver-specific ChREBP-KO, and WT mice were maintained for 4, 12, and 36 weeks after a single application of DEN to induce the development of HCC.

In alignment with previous studies [18,25], systemic ChREBP-KO significantly prevented body weight gain compared to liver-ChREBP-KO and WT mice, indicating that the systemic loss of ChREBP affects energy metabolism and appetite regulation. It has been previously reported that the markedly reduced food intake in diabetic mice with a systemic ChREBP deficiency is accompanied by a decreased expression of the appetite-stimulating neuropeptide agouti-related protein (AgRP) [25]. Another explanation for the significantly reduced weight gain observed in systemic ChREBP-KO mice is an increase in thermogenesis. In brown adipose tissue, the mitochondrial uncoupling protein 1 (UCP1), a key mediator of thermogenesis, is upregulated under conditions of systemic ChREBP-knockout [26]. Correspondingly, ChREBPβ has been described as a negative regulator of thermogenesis in brown adipocytes, based on overexpression experiments in mice [27]. The resulting increase in energy expenditure, in combination with the reduced appetite, leads to lower weight gain compared to wildtype controls.

Additionally, our study describes that global ChREBP-deficient mice have an increased liver-to-body-weight ratio, which may have contributed to the development of hepatomegaly in these mice. The hepatomegaly in ChREBP-KO mice is mainly caused by an elevated liver glycogen content, rather than fat storage [25,28]. Moreover, ChREBP-KO mice exhibited significant glycogen accumulation in the nuclei of hepatocytes, especially after 4 weeks, which might be due to the immediate effects of the DEN administration. A conceivable cause of nuclear glycogen accumulation would be the reduced stability or increased permeability of the nuclear membrane due to DEN-induced reactive oxygen species (ROS) [6]. Previously, our lab reported that the systemic loss of ChREBP contributes to hepatic glycogen accumulation by the decreased activity of the gluconeogenic enzyme glucose-6-phosphatase [28].

The tendential decreasing proliferation activity with the increasing age indicates an age-related decline in the liver’s proliferative capacity as previously shown [29]. Furthermore, we observed that liver-specific ChREBP-KO mice exhibited a significantly lower proliferation in the liver parenchyma compared to other genotypes, as indicated by the Ki-67 labeling index. Consistently, no malignant transformation to HCC was observed in liver-specific ChREBP-KO mice after 36 weeks, despite the non-altered frequency of preneoplastic foci compared to WT mice. These findings not only support the oncogenic role of specific ChREBP isoforms in the HCC pathogenesis [21] but also provide evidence that the hepatic ChREBP deletion, while leaving tumor initiation unaffected, markedly reduces the tumor progression.

In contrast, the systemic ChREBP-KO led to the fewest preneoplastic foci after 36 weeks, while the tumor incidence was only slightly reduced compared to wildtype mice, and the hepatocyte proliferation remained unchanged. These findings suggest that the systemic ChREBP-KO appears to reduce the tumor initiation in particular, while the tumor progression is only slightly slowed.

These different findings in hepatocarcinogenesis could be caused by the different expression of ChREBP isoforms in the liver tissue of systemic ChREBP-KO and liver-specific ChREBP-KO mice. Hepatocytes of liver-ChREBP-KO mice still express ChREBPβ, as only exon 1 is removed, and ChREBPβ is transcribed from exon 4 onwards. In contrast, systemic ChREBP-KO mice are deficient for both isoforms. Based on our findings and previous results [24,30], it is likely that ChREBP isoforms have distinct functions. ChREBPα appears to primarily regulate glucose metabolism, whereas ChREBPβ rather regulates the de novo lipogenesis. Interestingly, the loss of ChREBPα alone in liver-ChREBP-KO mice seems more tumor-protective than the loss of both isoforms, suggesting that the enhanced glycolysis driven by ChREBPα particularly promotes tumor progression, while ChREBPβ may mediate tumor-suppressive effects. These tumor-suppressive functions of ChREBPβ could potentially involve the modulation of apoptosis or immune responses. Our findings highlight ChREBPα as a potential target for tumor-directed therapies. However, further studies are needed to clarify the organ- and metabolism-specific roles of both isoforms.

Metabolic alterations in tumor tissues provide a microenvironment for cell proliferation and survival, facilitating HCC progression [15]. Consistent with previous studies [31,32,33], our results also revealed that the upregulation of ACAC and FASN, key enzymes in de novo lipogenesis, facilitated HCC progression by meeting the increased energy demands of proliferating hepatocytes. Additionally, the altered glycolysis in HCC (so-called Warburg effect), as indicated by the elevated expression of HK-2 and PKM2, provides further evidence that the metabolic rewiring is a critical factor in sustaining the uncontrolled growth of HCC [34,35]. We also demonstrated less expression of p-AKT, p-mTOR, and p-4E-BP1 in ChREBP-KO mice compared to WT mice, indicating a ChREBP-associated regulation in AKT/mTOR pathways during HCC development [36]. Moreover, our results showed the reduced expression of p-ERK in ChREBP-KO mice, suggesting that ChREBP may also influence the activation of the Ras/MAPK signaling pathway. Emerging evidence revealed that the Ras/MAPK pathway is essential for regulating cell proliferation and survival and driving hepatocellular transformation and tumor progression [37]. Taken together, our results highlight the pathogenic role of ChREBP-associated metabolic pathways in promoting HCC growth and progression.

## 4. Materials and Methods

### 4.1. Animals

In this study, all animals received humane care according to the criteria outlined in the “Guide for the Care and Use of Laboratory Animals”, prepared by the National Academy of Sciences and published by the National Institutes of Health (NIH publication 86-23 revised 1985). The animal experiments were approved by the Animal Policy and Welfare Committee of the Universitaetsmedizin Greifswald, Germany (LALLF-MV, Rostock, Germany, ref. no. 7221.3-1-027/20, 6 August 2020). Housing of the animals was in accordance with the guidelines of the Society for Laboratory Animal Service and the German Animal Protection Law.

Highly inbred 4-week-old male C57Bl/6J wildtype (WT, ChREBP^+/+^; Charles River Laboratories, Sulzfeld, Germany), systemic ChREBP-knockout (KO, ChREBP^−/−^; B6.129S6-Mlxipl^tm1Kuy^/J, #010537; Jackson Laboratory, Bar Harbor, ME, USA), and liver-ChREBP-knockout (L-KO, ChREBP^−/−^ only in hepatocytes) mice were assigned randomly to 18 groups (Table 2). For creating the liver-ChREBP-KO, floxed mice (ChREBP fl/fl in which loxP sites encircle exon 1 in the mlxipl gene; purchased and originally prepared by Chan, Center for Comparative Medicine, Baylor College of Medicine, One Baylor Plaza, Houston, TX, USA) were bred to a line expressing the Cre-recombinase specifically in the liver (Alb-cre; B6.Cg-Speer6-ps1^Tg(Alb-cre)21Mgn^/J, #003574; Jackson Laboratory, Bar Harbor, ME, USA). Their homozygous offspring in the 2nd generation contain the hepatocyte-specific ChREBP-KO (liver-ChREBP-KO). Examination of ChREBP expression in liver tissue by Western blot revealed that both ChREBPα and ChREBPβ isoforms are significantly downregulated in systemic ChREBP-KO. In contrast, in liver-ChREBP-KO mice, only the expression of ChREBPα is downregulated, whereas the glucose-independent ChREBPβ is still expressed by hepatocytes (Appendix A).

The experimental groups (DEN in Table 2) were injected intraperitoneally with carcinogenic DEN (dose 5 mg/kg body weight) at the age of 4 weeks. The control groups received an intraperitoneal administration of 0.9% sodium chloride at the same time. All animals were maintained on a normal chow diet and received water ad libitum (V1534-000, ssniff Spezialdiäten, Soest, Germany; Appendix A) for 4, 12, and 36 weeks.

Body weight and blood glucose levels were measured monthly until the day of sacrifice. Blood samples were taken from the tail vein and tested with a portable glucometer (Accu-Chek Performa, Roche Diabetes Care, Mannheim, Germany).

### 4.2. Genotyping

To ensure that the ChREBP-KO and liver-ChREBP-KO mice corresponded to the desired genotype, genotyping of each animal in these groups was required. The tissue for this came from ear punches, which were used to identify the mice. For DNA isolation, the ear punch was incubated with lysis buffer and 0.2 g/L proteinase K overnight at 55 °C. At the end of the lysis process, the proteinase K was inactivated by heating to 95 °C for 10 min.

PCR ChREBP-KO: In the presence of the WT allele, the mix of primers A and AB_2 forms a product of 550 bp. Under ChREBP-KO, primers B and AB amplify a 450 bp fragment.

PCR liver-ChREBP-KO: Detection of the loxP sequences on the ChREBP/Mlxipl gene with the primers Chrebpgeno2-31 and Chrebpgeno2-51. The WT allele produced a 210 bp fragment and the floxed allele a fragment of 358 bp. A triple primer mix was used to detect the Cre-genotype (AlbCre R, AlbCre Cre F, AlbCre WT F). This resulted in a 351 bp fragment for WT and a 180 bp and 390 bp fragment for homozygous mice. All three bands occurred in heterozygotes.

The sequences of all used PCR primers are presented in the Appendix A.

### 4.3. Tissue Processing

Animals were sacrificed after 4, 12, and 36 weeks under anesthesia (ketamine/xylazine 100/8.8 mg/kg body weight). Tissue was perfused with a solution of 0.5% procaine hydrochloride and 4% dextran in Ringer’s solution (pH 7.4) by cannulating the abdominal aorta with a 23-gauge needle. Up to 1.5 mL of arterial blood was collected for biochemical analysis.

All organs were removed under artificial exsanguination. The middle lobe of the liver was frozen in isopentane (2-methylbutane) cooled with liquid nitrogen. The removed tissue was fixed in formalin and embedded in paraffin. Serial sections with a thickness of 1–2 µm were used for hematoxylin and eosin (HE) staining as well as for the periodic acid-Schiff (PAS) reaction.

### 4.4. Immunohistochemistry

Then, 1–2 µm thick formalin-fixed paraffin-embedded serial liver sections were manually stained for acetyl-CoA carboxylase (ACAC), fatty acid synthase (FASN), hexokinase II (HK-2), phosphorylated eukaryotic translation initiation factor 4E binding protein 1 (p-4E-BP1), phosphorylated/activated AKT (p-AKT), phosphor-extracellular signal-regulated kinase 1/2 (p-ERK1/2), pyruvate kinase M2 (PKM2), and phosphorylated mechanistic target of rapamycin (p-mTOR). For antigen retrieval, a citrate buffer of pH 6.0 was used. Endogenous peroxidase was blocked with 1% hydrogen peroxide, and positive reactivity of primary antibodies was performed by the ZytoChem Plus (HRP) Broad Spectrum Kit (Zytomed Systems, Berlin, Germany) and DAB as the chromogen substrate (Dako, Glostrup, Denmark).

Ki-67 immunohistochemistry was executed using an automated immunostainer (Leica Biosystems, Wetzlar, Germany) and the previously mentioned DAB kit.

All primary antibodies with detailed information are listed in Appendix A. Protein expression in immunohistochemically stained sections was quantified using the method described by Crowe et al., implemented with the Fiji (ImageJ) software 2.9 [38].

### 4.5. Histological Evaluation

The histological examination was performed with a Leica type 301-371.010 microscope at 40 to 400× magnification. The Ki-67 labeling index (Ki-67-LI) was used to determine proliferation activity and corresponds to the percentage of Ki-67-positive cell nuclei of 2000 hepatocytes. Three representative fields of view were analyzed at 100× magnification. At the same time, glycogen-storing hepatocytes and nuclei were evaluated.

### 4.6. Protein Isolation

We used 10–20 mg of frozen liver tissue from both tumor and non-tumor areas for our analyses. Given the occasional occurrence of very small hepatocellular adenomas (HCAs), we cannot entirely exclude the possibility of microscopic tumor foci in the non-tumor tissue. This tissue was lysed in Pierce RIPA buffer containing Pierce protease and phosphatase inhibitor (both from Thermo Fisher Scientific, Waltham, MA, USA). Total protein was briefly extracted using a Bullet Blender with 0.5 mm zirconium oxide beads (both from Next Advance, Troy, NY, USA) in microcentrifuge tubes according to the manufacturer’s instructions. Homogenized samples were then centrifuged at 13,000 rpm for 10 min at 4 °C, and supernatant, containing total cellular proteins, was collected and quantified with the Pierce BCA Protein Assay Kit (Thermo Fisher Scientific, Waltham, MA, USA) using the FLUOstar Omega microplate reader (BMG LABTECH, Ortenberg, Germany).

### 4.7. Western Blot

For Western blotting, equal amounts of proteins were separated on an 8–12% SDS-PAGE and then transferred to an Immobilon-FL PVDF membrane (Merck, Darmstadt, Germany). After transfer, the proteins were visualized with reversible Ponceau-S staining (Sigma-Aldrich, St. Louis, MO, USA), followed by a destaining and a subsequent blocking of the membranes for 1 h at room temperature using 5% milk in TBST. After blocking, the membranes were incubated overnight at 4 °C with respective primary antibodies (Appendix A) in 2,5% milk in TBST. Unbound primary antibody was removed by washing the membranes three times with TBST at room temperature. Finally, the membranes were incubated with the fluorophore-labeled secondary antibodies IRDye 800CW anti-Mouse or anti-Rabbit and 680LT anti-Rabbit (926-32280/926-32211/926-68071; Dilution 1:10,000; LI-COR Biosciences, Lincoln, NE, USA) and scanned with an Odyssey CLx Imaging System (LI-COR Biosciences, Lincoln, NE, USA).

The Western blot results were quantified by densitometric analysis using ImageJ.

### 4.8. RNA Isolation and Quantitative PCR (qPCR)

Briefly, tissues were homogenized in RNAse/DNAse-free microcentrifuge tubes with RNA lysis buffer using the Bullet Blender as described in Section 4.6. For RNA isolation and purification, the Total RNA Purification Kit (Jena Biosciences, Jena, Germany) was used according to the manufacturer’s instructions.

Afterwards, cDNA was synthesized from purified RNA with the RevertAid First Strand cDNA Synthesis Kit (Thermo Fisher Scientific, Waltham, MA, USA) according to the manufacturer’s instructions. SYBR Green-based qPCR was performed in a Rotor-Gene 6000 real-time PCR cycler (Corbett Research, Oatley, NSW, Australia) using the SensiFAST SYBR No-ROX Kit (Meridian Bioscience, Cincinnati, OH, USA) and specific forward and reverse primers. Relative gene expression was normalized to the 18S rRNA housekeeping gene. Amplification was determined for technical triplicates of 6 animals per group as well as for a non-template control. The primer sequences are shown in Appendix A. The relative gene expression was determined by the comparative CT method, also referred to as the (2^−ΔΔCT^) method.

### 4.9. Statistical Analysis

All statistical analyses were performed with IBM SPSS Statistics 30 or GraphPad Prism 6.0. Quantitative data are expressed as mean ± standard error of the mean (SEM).

Differences in body weight, blood glucose level, glycogen storage, diameter of tumor, proliferative activity, and biochemical assays were assessed using the Student’s t-test of normally distributed data; otherwise, the Wilcoxon Mann–Whitney U test was applied. Normal distribution was tested using the Shapiro–Wilk test. Fisher’s exact test was performed for testing differences in tumor frequency.

Differences were considered significant if *** *p* < 0.001, ** *p* < 0.01, and * *p* < 0.05. “n.s.” indicates no significance.

## 5. Conclusions

In conclusion, our study sheds light on the molecular alterations driving HCC progression within the DEN-induced mouse hepatocarcinogenesis model and thereby unveils a pivotal role for the ChREBP in the initiation and advancement of HCC. We delineate distinct functions of the ChREBP isoforms, identifying ChREBPα as a potential oncogene and ChREBPβ as a possible tumor suppressor, though further targeted investigations are required to confirm these roles. By elucidating the interplay of ChREBP with metabolic reprogramming, insulin signaling, and critical pathways, such as AKT/mTOR and Ras/MAPK, our findings pave the way for innovative therapeutic strategies against HCC. Future research must evaluate these approaches to unlock their potential in clinical applications.

## Figures and Tables

**Figure 1 ijms-26-06932-f001:**
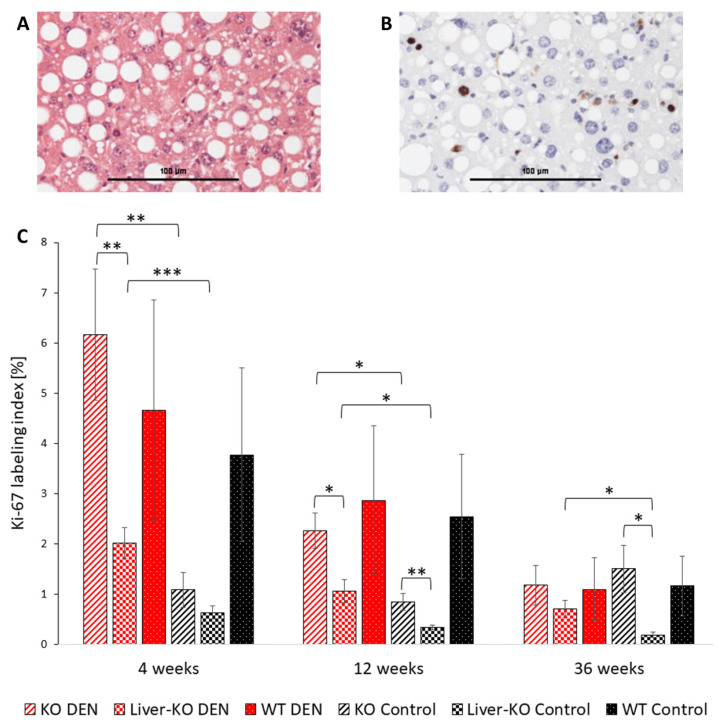
The proliferation activity of unaltered hepatocytes. (**A**) An exemplary micrograph of an H&E-stained liver section of a wildtype (WT) mouse 4 weeks after the diethylnitrosamine (DEN) application with slight fatty degeneration. (**B**) The immunohistochemistry staining of Ki-67 of the same section, ChREBP-KO DEN 4 weeks. Brown cell nuclei are in mitotic cell division and are counted to determine proliferation activity. The lower bar represents 100 μm. (**C**) The Ki-67 labeling index (percentage of Ki-67-positive hepatocyte cell nuclei) was measured in three randomly selected fields of ChREBP-KO, Liver-ChREBP-KO, and WT mice with DEN and the control after 4, 12, and 36 weeks (*n* = 309). Data are represented as the mean ± SEM. * *p* < 0.05; ** *p* < 0.01; and *** *p* < 0.001.

**Figure 2 ijms-26-06932-f002:**
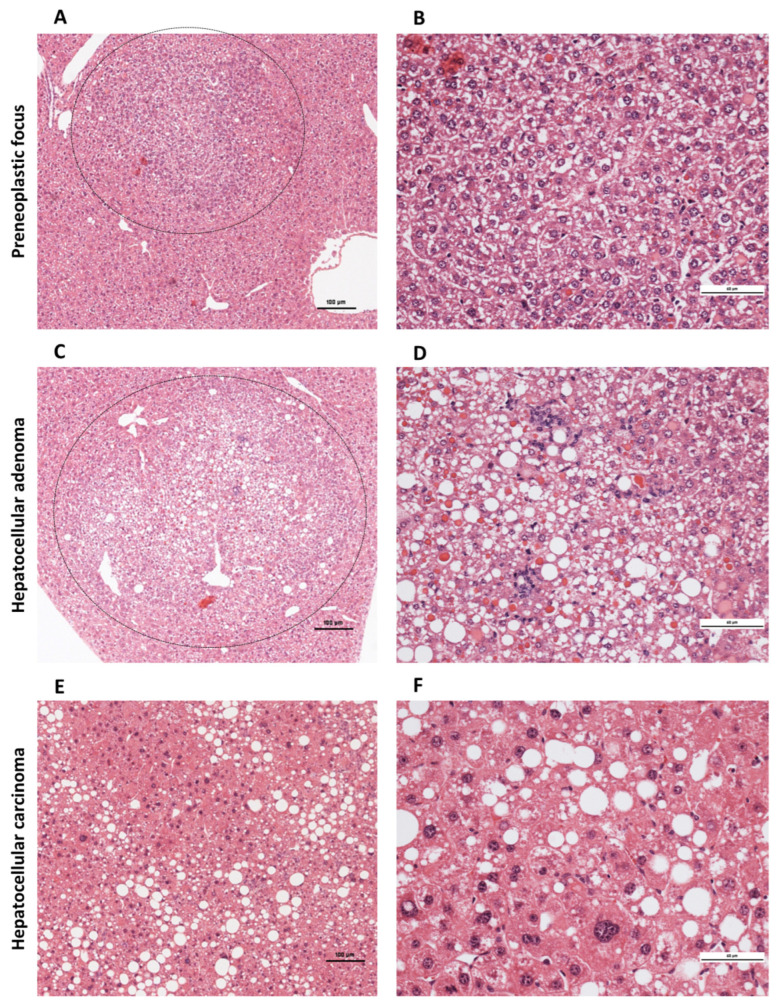
Preneoplastic lesions and tumors of WT mice 36 weeks after DEN administration. (**A**,**B**) Preneoplastic focus. (**C**,**D**) HCA. (**E**,**F**) Section of HCC shown in Figure 3 with severely dysplastic enlarged cell nuclei. Scale bars represent 100 μm (**A**,**C**,**E**) or 60 μm (**B**,**D**,**F**).

**Figure 3 ijms-26-06932-f003:**
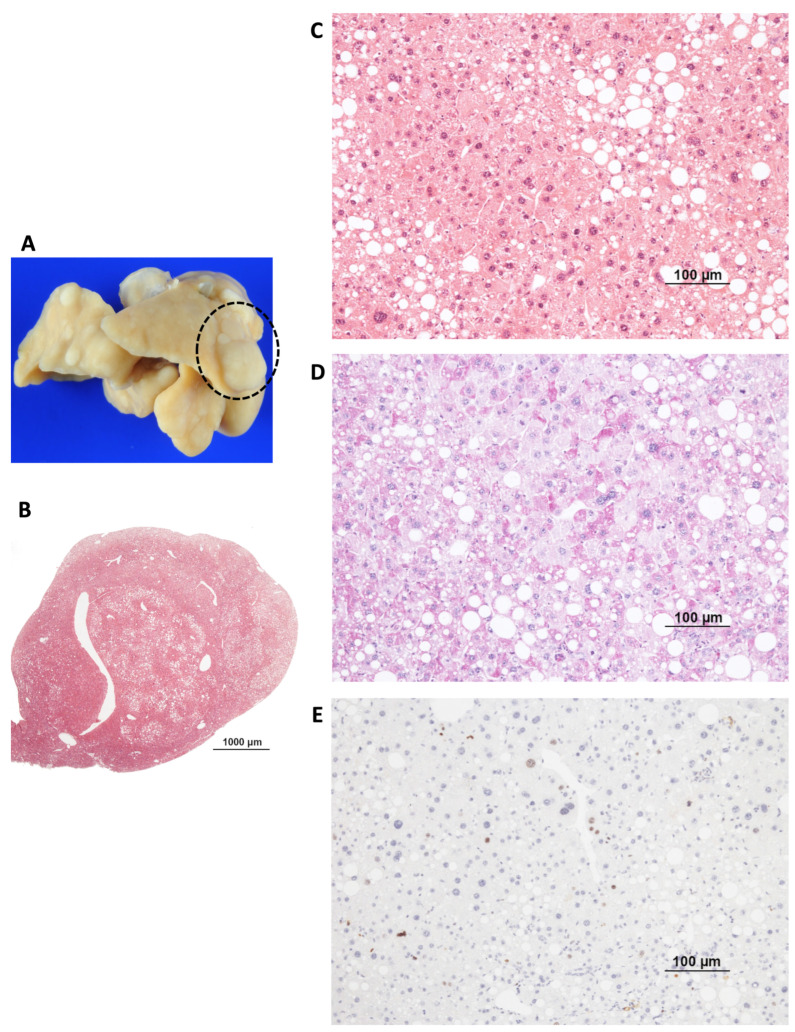
HCC of WT mouse after DEN administration and after 36 weeks. (**A**) Liver surface with many white–yellow nodules of different sizes and (**B**,**C**) HCC. (**D**) Deep-purple cytoplasm in PAS reaction due to massive glycogen storage in tumor cells. (**E**) Immunohistochemistry staining of Ki-67; brown cell nuclei are in mitotic cell division and reveal enhanced proliferation of tumor cells. The scale bar in image B represents 100 µm. Images (**C**–**E**) are on same scale, and bar represents 100 μm.

**Figure 4 ijms-26-06932-f004:**
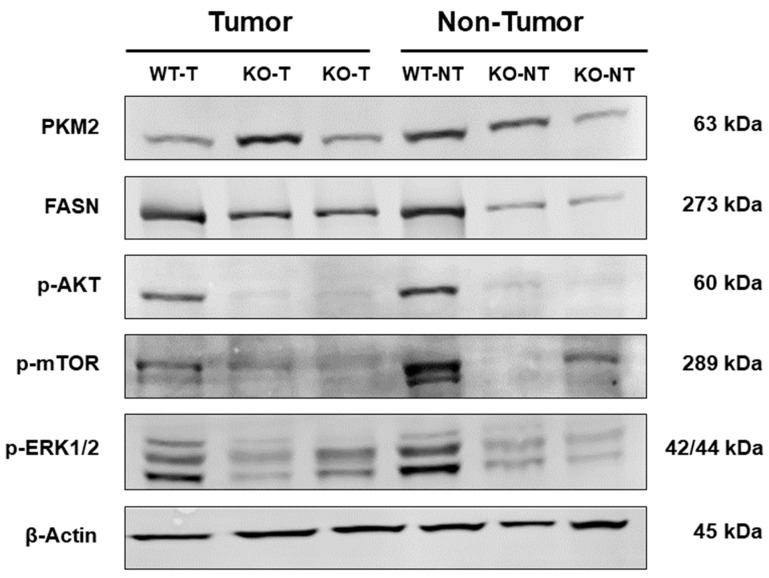
Protein expression in tumors and unaltered liver tissue of WT and ChREBP-KO mice 36 weeks after DEN administration. WT-T: Wildtype tumor after DEN application; KO-T: ChREBP-KO tumor after DEN application; WT-NT: Wildtype non-tumor tissue after DEN application; and KO-NT: ChREBP-KO non-tumor tissue after DEN application.

**Figure 5 ijms-26-06932-f005:**
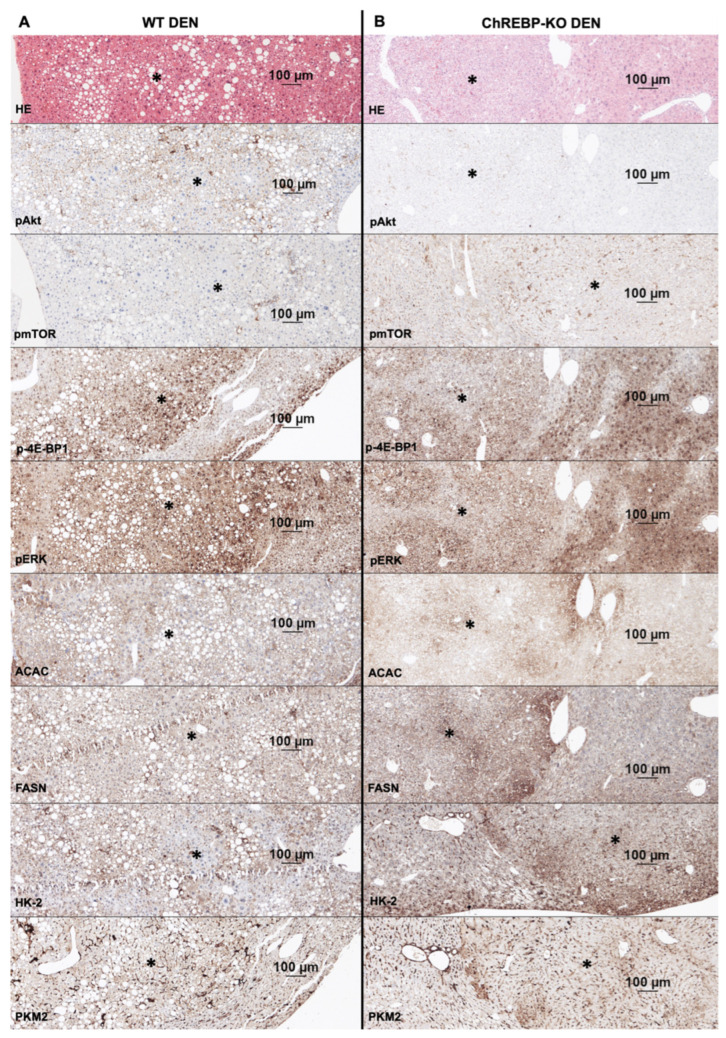
Representative immunohistochemical findings in liver tumors of WT and ChREBP-KO mice 36 weeks after DEN administration. (**A**) Representative micrographs of paraffin sections with WT tumor tissue (marked by *) and adjacent liver tissue. (**B**) Representative micrographs of paraffin sections with ChREBP-KO tumor tissue (marked by *) and adjacent liver tissue. All images are on same scale, and bar represents 100 μm.

**Table 1 ijms-26-06932-t001:** Frequency of preneoplastic lesions, hepatocellular adenomas (HCAs), and carcinomas (HCC) after 36 weeks.

	KO DEN	L-KO DEN	WT DEN	KO Control	L-KO Control	WT Control
n	22	22	23	25	24	25
Preneoplastic foci (%)	9.09 * #	39.13 * § #	40.91 §	8.33	0 §	8.00 §
HCA (%)	4.55 #	8.7	27.27 # §	8.33	0	4.00 §
HCC (%)	13.64	0	9.09	8.33	0	12
Tumors (HCA + HCC) (%)	18.18	8.7	27.27 §	8.33	0	12

* KO vs. L-KO; # KO vs. WT; § DEN vs. Control; and *p* < 0.05.

**Table 2 ijms-26-06932-t002:** Experimental and control groups.

	4 Weeks		12 Weeks		36 Weeks	
	DEN	Control	DEN	Control	DEN	Control
C57Bl/6J WT	*n* = 16	*n* = 24	*n* = 8	*n* = 24	*n* = 22	*n* = 25
ChREBP-KO	*n* = 17	*n* = 20	*n* = 8	*n* = 25	*n* = 22	*n* = 24
Liver-ChREBP-KO	*n* = 17	*n* = 25	*n* = 8	*n* = 25	*n* = 23	*n* = 25

## Data Availability

All results generated or analyzed during the present study are included in this published article and Appendix A. Data and materials will be made available upon request via an email to the corresponding author.

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
