# Peer review of "Carbohydrate-Responsive Element-Binding Protein-Associated Metabolic Changes in Chemically Induced Hepatocarcinogenesis Mouse Model"

_ijms, 2025, doi:10.3390/ijms26146932_

Round 1

Reviewer 1 Report

Comments and Suggestions for Authors

In the submitted manuscript, the authors propose to investigate the role of carbohydrate response element binding protein (ChREBP), a crucial regulator of hepatic glucose and lipid metabolism, in a chemical model of hepatocarcinogenesis.

The authors have already reported that hepatic ChREBP deletion ameliorates hepatic inflammation and impairs hepatocarcinogenesis in the high-fat diet (HFD)-induced mouse model.

The paper is well-written and the study is well-designed. However, there are some points that should be deepened.

  1. First of all, the authors should analyze the levels of Cyp2e1, involved in the metabolic bioactivation of diethylnitrosamine, in ChREBP-knockout (KO), hepatocyte-specific ChREBP-KO (L-KO), and wildtype (WT) mice.
  2. The authors should explain the choice of DENA dose in their study.
  3. What are plasma ALT levels in WT, KO and L-KO mice following DENA administration?
  4. Fig 1 B and C. Are there any significant differences in the Ki-67 labeling in non-parenchymal liver cells? What is the apoptotic index in these animals?

Author Response

Comments 1: First of all, the authors should analyze the levels of Cyp2e1, involved in the metabolic bioactivation of diethylnitrosamine, in ChREBP-knockout (KO), hepatocyte-specific ChREBP-KO (L-KO), and wildtype (WT) mice.

Response 1: We thank the reviewer for the suggestion to analyze Cyp2e1 levels, as CYP2E1 plays a central role in the metabolic activation of diethylnitrosamine (DEN). However, we did not include this analysis in our study, as we consider it unlikely to provide additional mechanistic insight into our main research question. The biological effects of DEN were clearly evident in all genotypes, as reflected by the distinct differences in tumor initiation and progression. This indicates that DEN was effectively metabolized across all groups, and suggests that the observed phenotypes are due to the specific roles of the ChREBP isoforms rather than differences in DEN activation.

This is supported by the study of Kang et al. (2007), who showed that Cyp2e1-knockout mice exhibited a marked reduction in DEN-induced liver tumorigenesis compared to wild-type mice, confirming the essential role of CYP2E1 in DEN activation (DOI: 10.1158/0008-5472.CAN-07-2172). Since our model does not involve direct modulation of Cyp2e1 expression or function, we believe that additional analysis of this enzyme would not substantially contribute to the interpretation of our findings.

Comments 2: The authors should explain the choice of DENA dose in their study.

Response 2: We used a single intraperitoneal injection of 5 mg/kg body weight of diethylnitrosamine (DEN) at 4 weeks of age, a dose commonly applied in genetic mouse models with increased susceptibility to hepatocarcinogenesis. This relatively low dose was chosen to enable the detection of genotype-dependent differences in tumor initiation and progression without inducing excessive liver damage or overwhelming tumor burden. Our aim was to investigate the specific contribution of ChREBP isoforms under moderate carcinogenic pressure. This approach is supported by previous studies, including Heindryckx et al. (2009), who demonstrated that low-dose DEN administration allows for mechanistic studies of hepatocarcinogenesis in transgenic models (DOI: 10.1007/s00432-007-0336-4).

Comments 3: What are plasma ALT levels in WT, KO and L-KO mice following DENA administration?

Response 3: We thank the reviewer for this valuable question. Plasma ALT levels are indeed an important marker of hepatocellular injury. However, we did not include serum ALT measurements in this study, as our experimental design was focused on long-term tumor development and hepatocyte proliferation rather than acute liver injury following DEN administration.

While we agree that ALT analysis could provide additional insight, especially regarding early hepatotoxic effects of DEN, we were not able to perform these measurements within the 10-day revision period. The ALT assay kit routinely used in our lab (Merck) was unexpectedly unavailable, and alternative kits from other providers indicated delivery times of 7 to 16 working days. This made it unfeasible to include these data in the current revision.

We appreciate the reviewer’s suggestion and consider serum transaminase measurements an important addition for future studies focused on acute liver injury and early DEN response in the context of ChREBP isoform function.

Comments 4: Fig 1 B and C. Are there any significant differences in the Ki-67 labeling in non-parenchymal liver cells? What is the apoptotic index in these animals?

Response 4: We thank the reviewer for this important question. In our study, we quantified the Ki-67 labeling index exclusively in hepatocytes to assess proliferation in the context of hepatocarcinogenesis. Proliferation of non-parenchymal liver cells, such as infiltrating immune cells or hepatic stellate cells , was not evaluated. Similarly, we did not assess apoptotic activity or determine an apoptotic index.

Our focus was on the role of ChREBP isoforms in liver tumor development and the corresponding proliferative response in non-tumorous liver tissue. An extended analysis including non-parenchymal cell proliferation or apoptotic markers would indeed provide valuable complementary information but was beyond the scope and aim of the present study.

Additionally, the timeframe for the current revision was limited to 10 days, which does not allow for the inclusion of further experiments of this kind. We nonetheless appreciate the reviewer’s suggestion and consider this a valuable avenue for future investigations.

Reviewer 2 Report

Comments and Suggestions for Authors

The manuscript suggests a potential tumor-suppressive role for ChREBPβ, but this is not deeply explored. Please consider adding a brief discussion on possible mechanisms or supporting evidence, even if speculative, to strengthen this point.

The manuscript is generally well written, but there are minor grammatical and phrasing issues. For example, the phrase “underscores the plausible oncogenic role of ChREBPα” (line 24) could be made clearer. A careful proofreading is recommended.

Author Response

Comments 1: The manuscript suggests a potential tumor-suppressive role for ChREBPβ, but this is not deeply explored. Please consider adding a brief discussion on possible mechanisms or supporting evidence, even if speculative, to strengthen this point.

Response 1: We are currently investigating apoptosis and immune response as possible mechanisms for tumor suppressive effects of the isoform ChREBPβ. With regard to this, we have adapted the discussion under line 296-310 accordingly: "These different findings in hepatocarcinogenesis could be caused by the different expression of ChREBP isoforms in the liver tissue of systemic ChREBP-KO and liver-specific ChREBP-KO mice. Hepatocytes of liver-ChREBP-KO mice still express ChREBPβ, as only exon 1 is removed and ChREBPβ is transcribed from exon 4 onwards. In contrast, systemic ChREBP-KO mice are deficient for both isoforms. Based on our and previous results [24,30], it is likely that ChREBP isoforms have distinct functions. ChREBPα appears to primarily regulate glucose metabolism, whereas ChREBPβ rather regulates the de novo lipogenesis. Interestingly, the loss of ChREBPα alone in liver-ChREBP-KO mice seems more tumor-protective than the loss of both isoforms, suggesting that enhanced glycolysis driven by ChREBPα particularly promotes tumor progression, while ChREBPβ may mediate tumor-suppressive effects. These tumor-suppressive functions of ChREBPβ could potentially involve modulation of apoptosis or immune responses. Our findings highlight ChREBPα as a potential target for tumor-directed therapies. However, further studies are needed to clarify the organ- and metabolism-specific roles of both isoforms."

Comments 2: The manuscript is generally well written, but there are minor grammatical and phrasing issues. For example, the phrase “underscores the plausible oncogenic role of ChREBPα” (line 24) could be made clearer. A careful proofreading is recommended.

Response 2: Thank you for pointing this out. We have amended the corresponding sentence in lines 22-26 as follows: "Our results showed that liver-specific loss of ChREBPα, while ChREBPβ remained active, significantly reduced tumor progression, suggesting an oncogenic role for ChREBPα. In contrast, systemic knockout of both ChREBPα and ChREBPβ reduced tumor initiation but did slightly prevent tumor progression, indicating that ChREBPβ may exert tumor-suppressive functions."

The renewed proofreading revealed no additional grammatical or phrasing issues.

Round 2

Reviewer 1 Report

Comments and Suggestions for Authors

No additional comments.